# Low-noise optomechanical single phonon-photon conversion for quantum networks

Liu Chen [1,5], Alexander Rolf Korsch [1,2,3,5], Cauê Moreno Kersul [4], Rodrigo Benevides[4], Yong Yu [1], Thiago P. Mayer Alegre [4] & Simon Gröblacher [1] ✉

Nano-structured optomechanical crystals (OMC) form an interface between mechanical modes with long coherence times and telecom optical photons, ideal for long-distance distribution of quantum information. However, the implementation of scalable quantum networks based on OMCs has been inhibited by thermal mechanical noise. Here, we overcome this limitation using a quasi-two-dimensional OMC and generate single photons via single phonon-photon conversion. In this work, we verify the low thermal noise and high purity of the generated single photons through a Hanbury Brown-Twiss experiment with $g^{(2)}(0) = 0.35^{+0.10}_{-0.08}$. We perform Hong-Ou-Mandel interference of the emitted photons showcasing the indistinguishability and coherence with visibility $V = 0.52 \pm 0.15$ after 1.43 km fiber delay. Lastly, we use two-photon interference to measure the temporal wavepackets of optomechanically generated single photons demonstrating narrow bandwidths as low as 10 MHz. Our results pave the way for multinode quantum networks of mechanical oscillators and hybrid entanglement generation between mechanical oscillators and telecom quantum emitters.

Quantum networks, used to distribute quantum information over long distances between many physical nodes, hold great promise for the realization of networked quantum computation, quantum communication, as well as distributed quantum sensing[1]. A high-fidelity interface between a long-lived quantum memory and optical photons for long-distance distribution of entanglement forms the fundamental building block of any practical quantum network[2,3]. Such light-matter interfaces have been realized in various physical platforms ranging from atomic vapors[4,5], trapped ions[6], individual trapped atoms[7] to color center defects in solid-state crystals[8,9]. Alternatively, entangled photon pairs from spontaneous parametric downconversion sources can be stored in the collective excitations of rare-earth ion doped crystals to distribute entanglement[10,11].

In recent years, integrated optomechanical crystals (OMCs) have emerged as a promising and versatile platform for quantum technologies[12–14]. The flexible operating wavelength of OMCs – including the telecom C-band where fiber transmission losses are minimized – as well as the long lifetimes of their mechanical mode of up to $T_1 \approx 1\,\text{s}$[15] and coherence times $T_2^* \approx 100\,\mu\text{s}$[16] make OMCs a natural candidate for storage and distribution of quantum information in long-distance quantum networks (see Fig. 1a). Furthermore, phonons in the mechanical mode can be readily coupled to other quantum systems such as solid-state defects[17,18], quantum dots[19], and most prominently superconducting circuits[20–24], enabling hybrid quantum information architectures for quantum information processing and networked quantum computation[25].

Using a heralded entanglement scheme based on the Duan-Lukin-Cirac-Zoller (DLCZ) protocol[26], entanglement between the mechanical modes of two OMCs has been demonstrated[27] and such entanglement has been used as a resource to perform an optomechanical Bell test[28]

[1]Kavli Institute of Nanoscience, Department of Quantum Nanoscience, Delft University of Technology, Delft, The Netherlands. [2]Department of Physics, Fudan University, Shanghai, China. [3]Department of Physics, School of Science, Westlake University, Hangzhou, China. [4]Instituto de Física Gleb Wataghin, Universidade Estadual de Campinas (UNICAMP), Campinas, SP, Brazil. [5]These authors contributed equally: Liu Chen, Alexander Rolf Korsch. ✉e-mail: s.groeblacher@tudelft.nl

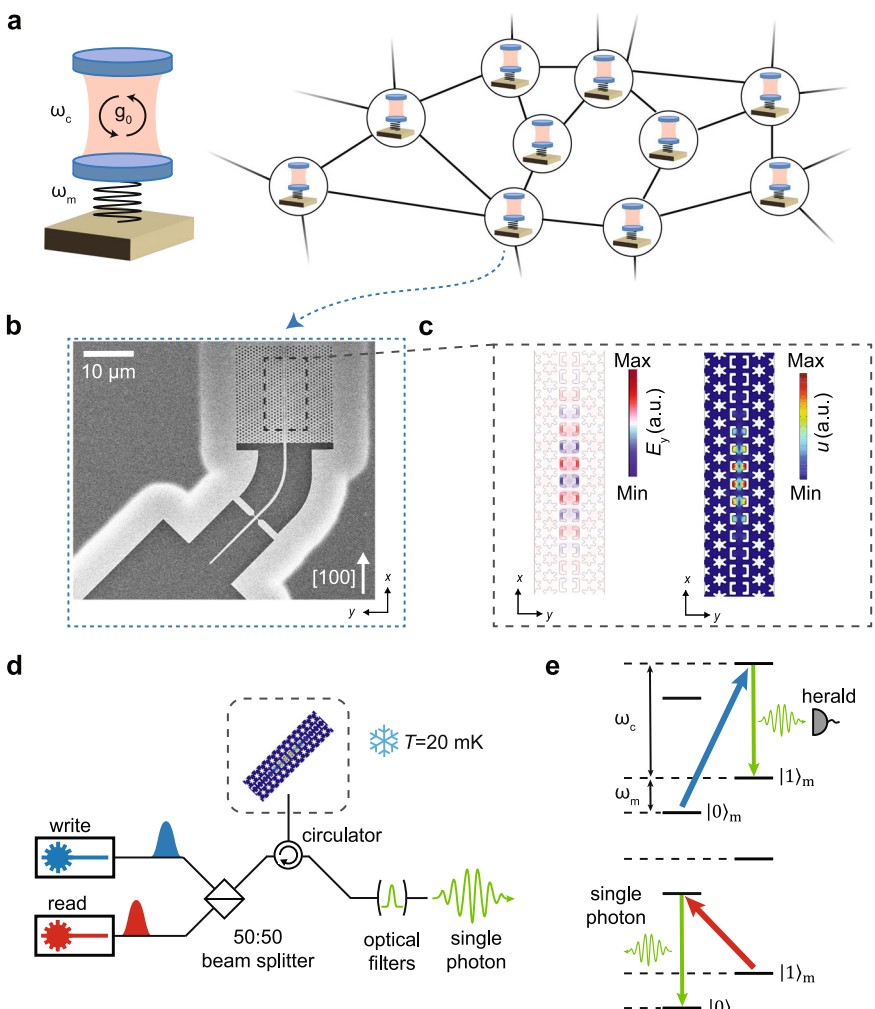

**Fig. 1 | Quantum network based on optomechanical crystals. a** Schematic illustration of a quantum network consisting of cavity optomechanical systems as network nodes with optical cavity frequency $\omega_c$, mechanical frequency $\omega_m$, and single-photon optomechanical coupling strength $g_0$. **b** Scanning electron microscope image of a 2D OMC device, which can form one of the telecom-wavelength quantum network nodes. The principal axis of the OMC cavity is aligned to the [100] direction of the silicon crystal lattice. **c** FEM simulations of the electric field (left) of the optical mode $E_y$ at design wavelength $\lambda$= 1537.24nm and the displacement field (right) of the mechanical mode at design frequency $\omega_m/2\pi$ = 10.18GHz. **d** Schematic illustration of the optical measurement setup. Write (read) laser pulses detuned to the blue (red) optomechanical sideband are sent to the OMC device inside a dilution refrigerator at base temperature $T$ = 20mK via lensed fiber

coupling. Single photons created through optomechanical interaction are filtered from the reflected light using optical filters locked to the optical cavity resonance of the OMC. **e** Illustration of the optomechanical Stokes- and anti-Stokes scattering processes used for single-phonon generation and phonon-photon conversion. Top: photons from the blue-detuned write pulse at frequency $\omega_b = \omega_c + \omega_m$ undergo a Stokes scattering process resulting in the probabilistic creation of a single photon at the optical cavity frequency $\omega_c$ and a single phonon at the mechanical frequency $\omega_m$. Detection of a single photon heralds the preparation of the mechanical mode in the Fock state $|1\rangle_m$. Bottom: the red-detuned readout pulse at frequency $\omega_r = \omega_c - \omega_m$ induces an anti-Stokes scattering process converting the single phonon in the mechanical mode to a single photon in the telecom band.

as well as optomechanical quantum teleportation[29] – demonstrating crucial steps towards quantum networks of mechanical oscillators. However, these initial demonstrations using one-dimensional nanobeam OMCs suffered from significant thermal noise due to optical absorption heating and weak thermal anchoring to the substrate[30], resulting in a relatively low purity of the optomechanically generated single photons[31], and thus inhibiting scaling up of the optomechanical DLCZ scheme to more complex quantum networks. To address this issue, quasi-two-dimensional (2D) OMCs have been developed, allowing for more efficient dissipation of generated thermal phonons into the cryogenic environment[32–34]. The improved thermal noise performance of such 2D OMCs has been verified in previous studies by measuring the thermal phonon occupancy of the mechanical mode[35,36]. However, to use such devices in quantum information distribution as well as networked quantum computation, it is crucial to demonstrate the purity and coherence of optomechanically generated

single photons. In particular, although previous experiments on entanglement of two mechanical oscillators imply the indistinguishability of optomechanically generated single photons[27], a direct demonstration of photon indistinguishability by two-photon interference has so far remained elusive due to prohibitively low optomechanical scattering rates limited by thermal noise.

In this work, we demonstrate heralded single-phonon generation and single phonon-photon conversion to telecom single-photons using integrated 2D OMCs with low thermal noise compatible with entanglement and quantum repeater schemes following the DLCZ protocol. We verify the low thermal noise of our device and demonstrate strong and robust quantum correlations between phonons and photons even at high mechanics-to-optics conversion efficiencies of up to 58%. We characterize the single-photon purity of the generated state through a Hanbury Brown-Twiss experiment. The obtained value $g^{(2)}(0) = 0.35^{+0.10}_{-0.08}$ is the lowest measured for integrated OMC systems

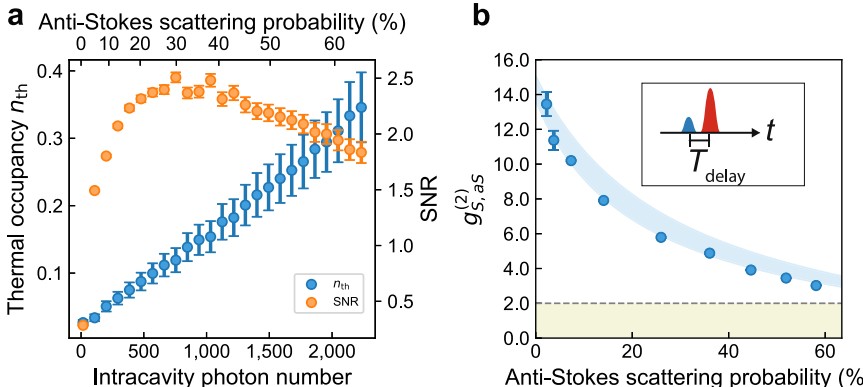

**Fig. 2 | Thermal performance and quantum cross-correlations. a** The blue dots show the thermal phonon occupancy $n_{th}$ of the mechanical mode as a function of intracavity photon number (bottom) and anti-Stokes scattering probability (top). Error bars originate from errors in the calibration of the detection path efficiency (see Supplementary Information). The orange dots indicate the signal-to-noise (SNR) ratio in the conversion process. **b** Cross-correlation function $g_{S,aS}^{(2)}$ between optomechanically scattered photons from the write and read pulse as a function of anti-Stokes scattering probability of the readout pulse. The insert shows the pulse sequence used for the cross-correlation measurement. A blue-detuned write pulse creates a single phonon, which is read out by a red-detuned read pulse after a delay

time of $T_{delay} = 150$ ns. The pulse sequence is repeated with a repetition period of $T_{rep} = 10\,\mu s$. The Stokes scattering probability of the write pulse is fixed at $p_S = 1.3\%$. The blue shaded area corresponds to the theoretically expected dependence $g_{S,aS}^{(2)} = 1 + e^{-T_{delay}/\tau_m}/(p_S + n_{th})$ (see Supplementary Information), where $n_{th}$ is calibrated from the results in **a** and $\tau_m = 1.0\,\mu s$ is the phonon lifetime of the mechanical mode (see Supplementary Information). The dashed horizontal line and shaded area underneath indicate the regime of classical correlations $g_{S,aS}^{(2)} \leq 2$. The error bars are calculated from the photon counting statistics and correspond to the 68% confidence interval of the binomial distribution.

and violates the threshold of $g^{(2)}(0) = 0.5$ for a genuine single-photon Fock state. The reduced thermal noise of our device enables operation at higher optomechanical scattering rates, allowing us to perform Hong-Ou-Mandel interference of subsequently emitted photons to quantify the coherence and indistinguishability of the generated single photons with a two-photon interference visibility of $V = 0.52 \pm 0.15$. Lastly, we use two-photon interference to measure the temporal wavepacket envelope of the emitted photons, demonstrating narrow linewidths as low as 10 MHz. The significant performance increase of 2D compared to 1D OMCs positions them as a promising platform to establish quantum networks of multiple mechanical oscillators. In addition, the narrow photon bandwidth as well as the (freely) designable operation wavelength make our system directly suitable for integration with narrow-linewidth telecom quantum emitters such as rare-earth ions[37–39] or silicon T-centers[40,41], as well as telecom quantum memories based on ensembles of rare-earth ions[42,43] or other optomechanical systems[44] in hybrid quantum networks.

## Results

### Single photon generation based on quasi-2D optomechanical crystals

Our device is a quasi two-dimensional suspended OMC structure (see Fig. 1b) fabricated on a silicon-on-insulator material platform based on a design of refs. 33,34. The two-dimensional anchoring allows efficient dissipation of thermal phonons generated through optical absorption heating. The optical and mechanical modes of our structure (see Fig. 1c) are confined along the $y$-direction by a snowflake crystal pattern exhibiting both photonic and phononic band gaps[32]. In between the snowflake areas, a periodic pattern of C-shaped holes modulated in size allows confinement along the $x$-direction. The $x$-direction of our device is oriented along the [100] crystal direction of the silicon lattice, which has been shown to lead to single-mode mechanical mode spectra robust against fabrication imperfections[34]. Co-localization of the optical and mechanical modes in a small mode volume leads to optomechanical coupling with a measured (simulated) single-photon coupling strength $g_0/2\pi = 1.0$ MHz ($g_{0,sim}/2\pi = 997$ kHz) (see Methods). Our device operates far in the resolved-sideband regime of cavity optomechanics where $\kappa \ll \omega_m$ with optical cavity linewidth $\kappa/2\pi = 2.4$ GHz and mechanical frequency $\omega_m/2\pi = 10.3699$ GHz.

We generate single photons through optomechanical heralded phonon generation and readout[13]. A laser pulse detuned on the blue optomechanical sideband of the optical cavity resonance induces a two-mode squeezing interaction between the optical and mechanical mode (see Fig. 1e). In the resulting Stokes scattering process, a single pump photon is probabilistically converted into a photon at the optical cavity resonance frequency and a phonon in the mechanical mode (see Fig. 1e). The light coming back from the device is filtered by a series of optical filters to remove the strong blue-detuned pump and detect single optomechanically scattered photons on superconducting nanowire single-photon detectors (SNSPD). Detection of the optical photon from the Stokes process heralds the creation of a single-phonon Fock state $|\psi_m\rangle \approx |1\rangle$. A second laser pulse detuned to the red optomechanical sideband transfers the phonon state onto the optical mode through a beam splitter interaction corresponding to an anti-Stokes scattering process (see Fig. 1e). This mechanics-to-optics conversion process allows us to generate single photons after filtering out the strong red pump pulse.

### Thermal phonon occupancy and quantum cross-correlations

To verify the improved thermal anchoring of our device compared to conventional one-dimensional nanobeam OMCs, we measure the thermal phonon occupancy of the mechanical mode $n_{th}$ for varying intracavity photon number $n_c$ in the optical cavity. We use 40-ns-long optical pulses detuned to the red optomechanical sideband and measure the count rate of optomechanically scattered anti-Stokes photons $C_{aS}$ after filtering out the strong pump pulse. By calibrating the efficiency of our detection path $\eta = 0.05$ and the anti-Stokes scattering probability $p_{aS}$ at various powers, we obtain the thermal phonon occupancy as $n_{th} = C_{aS}/(\eta p_{aS})$, where $p_{aS}$ is the anti-Stokes scattering probability at the respective pump power (see Methods). As shown in Fig. 2a, the thermal phonon occupancy increases with increasing intracavity photon number, but remains close to the quantum ground state even at photon numbers exceeding $n_c > 2000$, corresponding to a readout probability in the anti-Stokes scattering process of $p_{aS} > 60\%$. Compared to 1D structures, this represents a factor of three reduction of thermal occupancy of the mechanical mode at a similar level of intracavity photon number[29]. The highest anti-Stokes scattering probability achieved in these measurements is

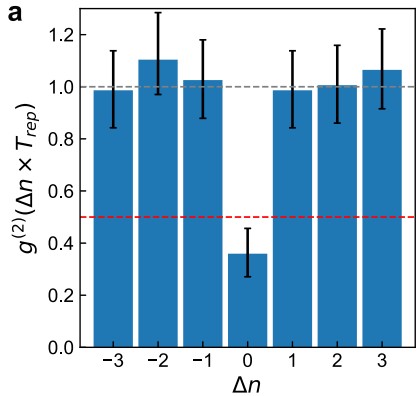
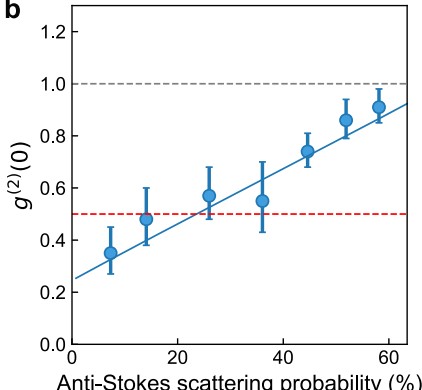

**Fig. 3 | Hanbury Brown-Twiss measurement. a** Measured second-order auto-correlation function $g^{(2)}$ of detection events from the read pulse conditioned on the detection of a Stokes-scattered photon during the blue-detuned write pulse. Each pulse sequence is labelled by a number $n$. Pulse sequences used for the $g^{(2)}$ calculation are shifted by $\Delta n$. The Stokes-scattering (anti-Stokes-scattering) probability are $p_S = 1.3\%$ ($p_{aS} = 7\%$) corresponding to pulse energies of $E_{p,S} = 0.1pJ$ ($E_{p,aS} = 1.0pJ$). **b** Second-order autocorrelation function $g^{(2)}(0)$ at fixed Stokes-scattering probability $p_S = 1.3\%$ as a function of anti-Stokes read out probability. Solid blue line

shows the result of simulations of the quantum systems using the Python package QuTIP (see Supplementary Information)[55,56]. A value below unity demonstrates sub-Poissonian photon statistics (grey dashed line in (**a** and **b**)) whereas a value below 0.5 unambiguously demonstrates a single-photon state (red dashed line in (**a** and **b**)). The error bars are calculated from the photon counting statistics using the exact binomial confidence interval (see Supplementary information). For all measurements, the optical pulse sequence is repeated with a repetition period of $T_{rep} = 10 \mu s$.

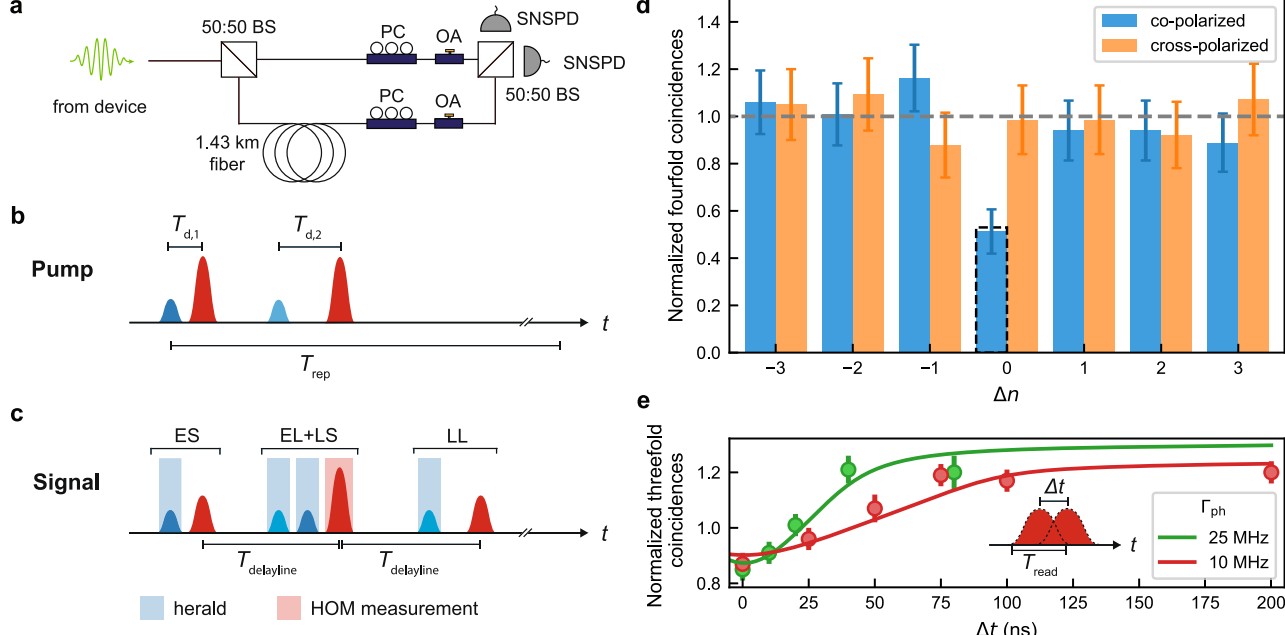

**Fig. 4 | Hong-Ou-Mandel interference. a** Unbalanced Mach-Zehnder interferometer used for Hong-Ou-Mandel (HOM) measurements. BS, beam splitter; PC, polarization controller; OA, optical attenuator; SNSPD, superconducting nanowire single-photon detector. **b** Pump pulse sequence used for HOM measurements. The delay between the blue and red pulses is $T_{d,1} = 105$ ns and $T_{d,2} = 225$ ns. The delay between the two red pulses equals the time delay induced by the fiber delay line with $T_{delayline} = 7.146 \mu s$. **c** Schematic of the measured detection events on SNSPDs. Time bins are labeled according to when the photon was created (E, early; L, late) and which interferometer arm it passed through (S, short; L, long). Photons generated from the first (second) blue-detuned pump pulses are shown in dark (light) blue. Simultaneous clicks during either combination of two dark and light blue-shaded time bins herald the generation of two phonons. The phonons are read out by the red pulses, leading to two-photon interference during the time bin

associated with the second red pulse (red shaded). **d** Number of four-fold coincidences measured on the two SNSPDs with co-polarized (blue) or cross-polarized (orange) arms of the interferometer during the same ($\Delta n = 0$) or different ($\Delta n \neq 0$) repetitions of the experiment, normalized to the average value measured on the satellite peaks ($\Delta n \neq 0$). The error bars are calculated from the photon counting statistics and correspond to the 68% confidence interval of the binomial distribution. The dashed bar at $\Delta n = 0$ shows the value predicted from QuTiP simulations (see Supplementary Information). **e** Normalized number of threefold coincidences in co-polarized interferometer configuration during the same repetition of the experiment ($\Delta n = 0$) as a function of timing offset $\Delta t$ between the two red pulses for two different bandwidths $\Gamma_{ph}$ of generated photons. The solid lines are fits to a phenomenological model based on the photon pulse shape (see Supplementary Information).

limited only by the power of our laser and residual losses in the optical setup. If we define the signal-to-noise ratio (SNR) as $\xi = p_{as}/n_{th}$, we obtain the highest SNR of $\xi = 2.5$ at about 30% of anti-Stokes scattering probabilities, a factor of 2.5 increase compared to the highest value achieved in 1D structures[29]. Based on the low thermal occupancy of our device, we demonstrate strong non-classical correlations between Stokes- and anti-Stokes-scattered photons detected on the SNSPDs during the write and read pulses[13]. The measured correlations (see Fig. 2b) exceed the classical limit by 51 standard deviations even at high anti-Stokes scattering probability up to $p_{aS} = 58\%$.

### Characterization of single photon purity

The purity of the generated single photons can be characterized by performing a Hanbury Brown-Twiss (HBT) experiment on the read-out optical state conditioned on the detection of a Stokes-scattered photon during the write pulse. Figure 3a shows the result of the HBT measurement for phonons read from the same ($\Delta n = 0$) or different ($\Delta n \neq 0$) repetition periods (see Supplementary Information). We observe strong anti-bunching of the read-out photons and determine a value of the conditional autocorrelation function of $g^{(2)}(0) = 0.35^{+0.10}_{-0.08}$, which is significantly below the limit of $g^{(2)}(0) = 0.5$ for a genuine single photon state[45]. The measured value of $g^{(2)}(0)$ is mainly limited by residual absorption heating from the write and read pulse, creating added thermal noise on the read-out optical state. Dark counts account for 0.78% of the total coincidences (see Supplementary Information). At higher anti-Stokes scattering probabilities, more thermal noise is added, reducing the fidelity of the readout optical state (see Fig. 3b). Nonetheless, we observe sub-Poissonian photon statistics with $g^{(2)}(0) < 1$ even for the highest anti-Stokes scattering probabilities used.

### Hong-Ou-Mandel interference of optomechanically generated single photons

Photon indistinguishability is crucial for many quantum information processing protocols, including building quantum repeaters for long-distance quantum networks. We verify the coherence and indistinguishability of the single photons generated by our source by performing Hong-Ou-Mandel (HOM) interference. We pass two subsequently generated single photons through an unbalanced Mach-Zehnder interferometer with a 1.43 km fiber delay line in one arm corresponding to a time delay of $T_{delayline} = 7.146\,\mu s$ (see Fig. 4a). To generate two subsequent photons, we use the pulse sequence shown in Fig. 4b. Within one repetition period ($T_{rep} = 18\,\mu s$), two pairs of blue and red detuned pulses (40 ns) are sent to the device to generate single phonons ($p_S = 10\%$) and read them out ($p_{aS} = 45\%$). The pulse energies of the write and readout pulses are $E_{p,S} = 1.1\,pJ$ and $E_{p,aS} = 8.3\,pJ$, respectively. The delay between the two red read-out pulses in the two pulse groups is set to be equal to $T_{delayline}$, leading to HOM interference at the second beam splitter (see Fig. 4c). The long time delay $T_{delayline} \gg \tau_m$ between subsequently generated photons allows the mechanical mode to thermalize to the cryogenic environment before each photon is generated. For the HOM measurement, we measure four-fold coincidences between two clicks from the blue write pulses and two clicks from the red read out pulses. When the two interferometer arms are co-polarized, the photons arriving at the beam splitter during the same repetition are indistinguishable resulting in the characteristic dip in coincidence detection events (see Fig. 4d). As a control experiment, we repeat the same measurement with cross-polarized arms of the interferometer observing no dip in coincidence events, as expected. From the two measurements, we calculate the HOM interference visibility $V_{raw} = 0.48 \pm 0.14$. After correcting the power imbalance of the two arms (see Supplementary Information), we obtain a HOM visibility of $V = 0.52 \pm 0.15$. The visibility is reduced compared to the case of ideal single photons ($V = 1$) due to the added thermal component of the optical state. We model the impact of the

thermal component on the HOM visibility through numerical simulations and determine a simulated visibility of $V_{sim} = 0.53$ (see Supplementary Information) in good agreement with the measured value. Although, the HOM visibility does not violate the theoretical bound for non-classical states $V > 0.5$, the observation of HOM interference nevertheless demonstrates the coherence and indistinguishability of optomechanically generated single-photons generated more than 7 $\mu s$ apart in time.

### Measurement of photon temporal wavepacket and bandwidth

Finally, we use two-photon interference to measure the temporal wavepacket shape and thus the bandwidth of optomechanically generated single-photons, offering insight for interfacing with other components (e.g., quantum memories) in quantum networks. For short readout pulse durations used in our experiment $T_{read} \ll 2\pi/\Gamma_{om}$, the photon bandwidth is expected to closely follow the readout pulse wavepacket shape[46,47]. Here, $T_{read}$ is the readout pulse length and $\Gamma_{om} = \Gamma_m + 4n_c g_0^2/\kappa$ is the optomechanically enhanced mechanical linewidth with the intrinsic mechanical linewidth $\Gamma_m$. The photon bandwidth $\Gamma_{ph}$ is given as the inverse of the temporal wavepacket shape of the generated photon. We verify the temporal wavepacket shape and hence the photon bandwidth by offsetting the two readout pulses by a time delay $\Delta t$, which reduces the overlap of two photons arriving at the beamsplitter, leading to reduced HOM interference. To increase the statistics, we measure three-fold coincidences between one click from the blue write pulses and two clicks from the red readout pulses. Hence, only one single photon is generated while the other red-detuned pulse reads out an unheralded mechanical thermal state. This HOM interference between a single photon and a thermal state leads to a reduced depth of the observed HOM dip. Figure 4e shows the normalized number of threefold coincidences as a function of timing offset $\Delta t$ for photon bandwidths of 25 MHz and 10 MHz, corresponding to read-out pulse lengths of 40 ns and 100 ns, respectively. The HOM dip persists up to longer time delays for longer readout pulse length, showcasing the tunability of the temporal shape of the photon wavepacket in agreement with a phenomenological model (solid lines). We note that as the pulse offset is increased, the normalized number of coincidences increases above the value of unity as a consequence of the bunched photon statistics of the thermal state (see Supplementary Information).

## Discussion

Our work demonstrates heralded single phonon creation and phonon-photon conversion at telecom wavelengths based on a 2D OMC platform, paving the way for building a quantum network based on nano-structured OMCs. In contrast to previous experiments, in which the measurement of the photon autocorrelation function was limited by thermal noise to greater than the threshold of $g^{(2)}(0) = 0.5$ of a genuine single-photon Fock state, the reduced thermal noise of our 2D OMC enables single-photon emission with $g^{(2)}(0) = 0.35^{+0.10}_{-0.08}$ in an HBT measurement, unambiguously demonstrating the single-quantum nature of the emitted state. Furthermore, the improved thermal performance enables operating the device at high optomechanical scattering probabilities and thus enables the realization of a HOM experiment, which requires the detection of four-photon coincidence events. This has so far proven elusive due to low rates caused by limited optomechanical scattering probability. Our experiment confirms the indistinguishable character of the generated photons through the observation of HOM interference with visibility $V = 0.52 \pm 0.15$, and by introducing a delay line for one photon of $T_{delayline} = 7.146\,\mu s$ further demonstrates that the emitted photons are coherent over this timescale. Since the typical amplitude of mechanical frequency jittering, which ultimately limits photon coherence $\Delta f_m \approx 10\,kHz$ is small compared to the emitted photon bandwidth, we expect the coherence of photons

emitted from our source to be preserved for long periods of time, ideal for long-distance quantum network applications.

Our 2D OMC device also showcases narrow photon linewidth as low as 10 MHz – essential for interfacing with telecom quantum memories, limited only by the optical pulse length used for phonon-photon conversion. While longer optical readout pulse lengths would directly enable narrower optical linewidth, ultimately only limited by the intrinsic mechanical linewidth $\Gamma_{\mathrm{m}}/2\pi = 119$ kHz (see Supplementary Information), this would lead to an accumulation of more thermal phonons in the mechanical mode[30] and thus currently still impede measurements in the quantum regime. Notably, compared to other systems that generate single photons at telecom C-band, the bandwidth of our system already surpasses the narrowest linewidths achieved with telecom quantum dots (>100 MHz[48]), as well as with silicon-based on-chip sources based on spontaneous four-wave mixing (≈30 MHz[49]). Moreover, since the temporal wavepacket shape of the emitted photons from our OMC device is determined by the pulse shape of the optical readout pulse, pulse shaping of the readout pulse can be used to generate shaped single photons. Shaped single-photons are essential to realize quantum interference with single photons from other quantum systems, such as rare-earth ions[37–39] or quantum dots[48].

While thermal noise currently still limits the single-photon purity and HOM visibility as well as the attainable narrowest optical linewidths, further improvements to the design to reduce pump-induced heating through the use of evanescently coupled optomechanical cavities or non-suspended OMCs exist[36,50], which will allow to further boost the purity and rate of generated single photons, enabling a more efficient quantum network of OMCs. The lifetime of the phonon mode in the device used in this work is chosen to be only $\tau_{\mathrm{m}} = 1$ μs to increase the repetition rate of the experiment. However, by engineering the phononic band structure of the OMC geometry, we have already fabricated devices with long phonon lifetimes up to $\tau_{\mathrm{m}} = 9$ ms (see Supplementary Information). This long phonon lifetime combined with high purity and long coherence time of optomechanically generated single photons at telecom wavelength, position 2D OMCs as a promising platform for the realization of long-distance quantum networks following the DLCZ protocol[26]. With our current performance, we estimate that heralded entanglement between two 2D OMCs embedded in a phase-stabilized Mach-Zehnder interferometer can be generated at a heralding rate of 100 Hz and verified at a total event rate of $2.9 \times 10^3$ h$^{-1}$ – an improvement by more than two orders of magnitude over previous demonstrations[27] (see Supplementary Information). The heralding rate is already surpassing demonstrations of heralded entanglement generation using nitrogen vacancy centers (39 Hz[51]). In a next step, well-established technical improvements of optical setup efficiency will further boost the attainable entanglement generation rates to 300 Hz, comparable with recent experiments based on atomic ensembles (280 Hz[52]). Importantly, the generated single-photons in[51] and[52] are, unlike for our device, not at telecom wavelength and hence for any practical quantum network application, wavelength conversion would reduce the attainable entanglement rate significantly. Further reduction of thermal noise by evanescent light coupling[36] will allow an additional boost of the entanglement rates to 1.2 kHz (see Supplementary Information), on par with leading quantum networking platforms based on cavity-enhanced parametric-down-conversion sources and atomic frequency comb quantum memories in solid-state crystals (1.4 kHz[11]), and in an on-chip platform compatible with scalable CMOS fabrication[53]. These rate improvements will enable the scale-up of these rudimentary quantum networks by demonstrating entanglement swapping between multiple pairs of entangled mechanical oscillators.

Finally, low-noise 2D OMCs are ideally suited to explore hybrid quantum network architectures interfacing different physical platforms: the narrow linewidth and by-design controllable wavelength of optomechanically generated single-photons allow for hybrid

entanglement creation through interference of single photons from an OMC and telecom quantum emitters, such as rare-earth ions[37–39] or silicon T-centers[40,41]. Alternatively, single-photons from OMC devices can be stored in telecom quantum memories with narrow acceptance bandwidth, such as those based on ensembles of rare-earth ions[42,43] or other optomechanical systems[44]. A second unique property of these systems is that phonons can be coupled directly to a large variety of other quantum systems, for example, to color centers[17,18] or quantum dots[19] through strain interaction. When combined with a piezo-mechanical element, high-efficiency microwave-to-optics conversion is also directly attainable, where the low thermal noise demonstrated in our device will enable coherent, low-noise on-demand entanglement between microwave superconducting quantum circuits, critical for future quantum operations involving distant quantum processors in networked quantum computing[20–24,54].

## Methods

### Nanofabrication
The quasi-2D optomechanical crystal structures are fabricated from a silicon-on-insulator wafer with a 250 nm device layer. Devices are patterned using electron beam lithography and HBr/Ar reactive ion etching. After etching, we perform cleaning using a piranha solution. The devices are released by wet etching of the buried oxide layer using hydrofluoric acid (40%).

### Calculation of $g_0$
We calibrate the vacuum optomechanical coupling rate $g_0$ by measuring the optomechanical scattering probabilities with calibrated input optical power and setup efficiency. We send either blue or red-detuned optical pulses with 40 ns pulse length to the device and measure the resulting count rates on the SNSPDs. The expected count rates for the Stokes and anti-Stokes process, $C_{\mathrm{S}}$ and $C_{\mathrm{aS}}$, are given by

$$C_{\mathrm{S}} = (\eta_1 + \eta_2) p_{\mathrm{S}} (1 + n), \tag{1}$$

$$C_{\mathrm{aS}} = (\eta_1 + \eta_2) p_{\mathrm{aS}} n, \tag{2}$$

where $\eta_1$ and $\eta_2$ are the efficiencies of the optical paths for SNSPD 1 or 2, respectively, and $n$ is the thermal phonon occupancy of the mechanical mode. The scattering probabilities for the Stokes and anti-Stokes process $p_{\mathrm{S}}$ and $p_{\mathrm{aS}}$ are given by[47]

$$p_{\mathrm{S}} = \exp\left[\frac{\kappa_{\mathrm{e}}}{\Delta^2 + \kappa^2/4} \frac{g_0^2 \kappa}{(\Delta - \omega_{\mathrm{m}})^2 + \kappa^2/4} N_{\mathrm{p}}\right] - 1, \tag{3}$$

$$p_{\mathrm{aS}} = 1 - \exp\left[-\frac{\kappa_{\mathrm{e}}}{\Delta^2 + \kappa^2/4} \frac{g_0^2 \kappa}{(\Delta + \omega_{\mathrm{m}})^2 + \kappa^2/4} N_{\mathrm{p}}\right], \tag{4}$$

where $\kappa$ and $\kappa_{\mathrm{e}}$ are the total optical linewidth and extrinsic optical coupling rate, and $N_{\mathrm{p}}$ is the number of photons in the excitation pulse.

For low optical power and laser pulses resonant with the blue or red optomechanical sideband ($\Delta = \pm \omega_{\mathrm{m}}$), we can approximate the exponential in the Stokes scattering probability in Eq. (1). Furthermore, at low optical powers the thermal phonon occupancy is small so that $n \ll 1$ and thus $n + 1 \approx 1$ in Eq. (1). We obtain for the vacuum optomechanical coupling rate

$$g_0 = \left(\frac{C_{\mathrm{S}}}{\eta_1 + \eta_2} \frac{\omega_{\mathrm{m}}^2 + \kappa^2/4}{4 N_{\mathrm{p}}} \frac{\kappa}{\kappa_{\mathrm{e}}}\right)^{1/2}. \tag{5}$$

From the measured count rate of the Stokes process $C_{\mathrm{S}}$, we obtain $g_0/2\pi = 1.0$ MHz.

**Measurement of the thermal phonon occupancy from the readout pulse**

We can determine the phonon occupancy $n$ by detuning the laser on the red sideband of the optical cavity and measuring the click rates of optomechanically scattered photons in the anti-Stokes process. From Eq. (2), we calculate the thermal phonon occupancy as

$$n = \frac{C_{aS}}{(\eta_1 + \eta_2)p_{aS}}. \tag{6}$$

For each laser power used in the experiment, we use the previously calculated value of $g_0$ and Eq. (4) to calculate the corresponding anti-Stokes scattering probability $p_{aS}$. We estimate the systematic error of the measured efficiency of the line to be 15%, which results in an increased error bar in the measured thermal occupancy $n_{th}$ as the measured thermal phonon number increases. This systematic error arises from fiber-optical connectors in our optical detection path, which need to be disconnected and reconnected to measure the path efficiency. The losses of each fiber optical connector vary each time a connector is reconnected. We calibrate this error by performing the same calibration of the detection path efficiency multiple times. From the standard deviation of the measured efficiency, we estimate the systematic error.

## Data availability
Source data for the plots are available on Zenodo.

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

## Acknowledgements

We would like to thank Radim Filip for helpful discussions. We further acknowledge assistance from the Kavli Nanolab Delft. This work is financially supported by the European Research Council (ERC CoG Q-ECHOS, 101001005) and is part of the research program of the Netherlands Organization for Scientific Research (NWO), supported by the NWO Frontiers of Nanoscience program, as well as through a Vrij Programma (680-92-18-04) grant. C.M.K., R.B., and T.P.M.A. acknowledge support by S ao Paulo Research Foundation (FAPESP) through grants 18/15580-6, 18/25339-4, 19/01402-1, 20/06348-2, 22/14273-8, and Coordenação de Aperfeiçoamento de Pessoal de Nível Superior - Brasil (CAPES) (Finance Code 001), and Financiadora de Estudos e Projetos (Finep).

## Author contributions

L.C., A.R.K., and S.G. devised and planned the experiment. C.M.K., L.C., and R.B. worked on the device simulation and design. L.C. and A.R.K. fabricated the device. C.M.K. assisted in the device characterization. L.C. and A.R.K. executed the experiments and performed data analysis with assistance from Y.Y. A.R.K., L.C., and S.G. wrote the manuscript with input from all authors. S.G. and T.P.M.A. supervised the project.

## Competing interests

The authors declare no competing interests.
