## [Transparent Peer Review file · Nature Communications]

Low-noise Optomechanical Single Phonon-photon Conversion for Quantum Networks

Corresponding Author: Professor Simon Groeblacher

Version 0:

Reviewer comments:

Reviewer #1

(Remarks to the Author)

The paper demonstrates the performance of an efficient photon-phonon converter based on a 2D optomechanical crystal device already demonstrated recently in Ref. 34 (see also Ref 35). Similar 1D devices of this kind have been already employed for early demonstration of single photon manipulation, quantum teleportation and entanglement, but the purity of the generated single photon pulses was not enough for an efficient use in a quantum network. In this paper, the significantly improved management of the heating caused by optical absorption enables a much more pure single photon and single phonon states. This is very clearly demonstrated through a Hanbury Brown-Twiss experiment and then also a Hong-Ou-Mandel interference experiment, achieving a performance which is the best one achieved with optomechanical devices, and approaching the one achievable with atomic ensembles. I agree with the authors that such on-chip devices could be easily integrated within hybrid platforms with quantum dots and superconducting qubits, and therefore are extremely promising. The data are clearly presented and discussed, and the supplementary materials provide a quite complete description of the techniques and methodology.

For the above reasons I recommend publication in Nature Communication. A suggestion for a possible improvement is related to the physics behind the thermal management, which has been partially discussed in the previous works (Refs 34 and 35). Apart from the intuitive argument based on the more efficient 2D versus 1D geometry, could the author provide some more detail or a model of why now optical absorption is less effective ? It is only a matter of better thermal conductance and dissipation ? Which is the effect of optical power on the mechanical and optical quality factors ? One could compare for example with the recent study of the mechanical quality factor versus T and optical power by F. Marzioni et al., Appl. Phys. Lett. 126, 174002 (2025), even though the latter refer to a 2D SiN membrane with a Si frame.

Reviewer #2

(Remarks to the Author)

The authors present an experiment that involves a two-dimensional optomechanical crystal (OMC) operated in the pulsed regime of single phonons. They provide evidence for the single phonon nature of their mechanical state by performing a g² (Hanbury Brown-Twiss) experiment, and evidence for the indistinguishability of these single phonon from one run to another by performing a Hong-Ou-Mandel experiment. The work is excellently written and referenced, and the execution of the experiment is of the highest quality.

Two-dimensional adaptations of the original one-dimensional OMC is a crucial upgrade of this experimental platform, that may enable deeply quantum operations without the common obstacle of thermal heating originating from the control optics. The idea of extending the OMC to a two-dimensional configuration is not novel, and promising results have been already published recently by other groups. The focus of the authors on the single phonon nature and on the indistinguishability, however, elevates the novelty of the work and convinced me that it deserves publication in Nature Communication, which I recommend.

However, I would ask the authors to provide few clarifications of their work first.

-- For readers less familiar with fabrication intricacies, could the authors comment a bit more about the specific configuration of their device? Would it be possible to achieve the same effect with different unit cells instead of their stars and C shapes? Clearly yes, given the work in Nat. Comm. 16, 2576 (2025), so I wonder what are the advantages and disadvantages of

different choices. Also, why is the waveguide that delivers light to the device curved? Is there something relevant on a physical level in such layout, or is it just for convenience?

-- The thermal improvement reported by the authors is a bit lower than other two-dimensional OMC experiments, could you estimate why is this the case? Was there a thermal analysis during the simulation stage?

-- In the experimental setup, the authors seem to be using two different lasers for the red and for the blue sidebands. What are actually the powers involved in this experiment? Could they not have executed the same experiment with only one laser and some kind of single sideband modulator?

-- Along the same line, I would have normally thought that the frequency of the lasers needs to be stabilized by looking at their reflection from the optical resonator of the OMC. The authors instead seem to be using only a wavemeter, which makes their stabilization open-loop with respect to their device under test. Is this possible because of the relatively large linewidth of their resonator, or is the device really that stable?

-- The authors report two devices with vastly different lifetimes, 1us against 9ms. Was this result on purpose or accidental?

-- What role does the length and shape of your pulse play? Is there anything to gain from shaping it to a more tailored envelope, for instance?

-- In the pump and probe experiment of the Supplemental, the authors mention that their probe pulse is 5us long, and that it is shone on the SNSPDs. Why is the duration of this pulse so long if the temporal dynamics they try to resolve is on the order of few hundreds of nanoseconds (Fig. S4)?

-- Why did the authors remove the polarization controllers before and after the device for the Hong-Ou-Mandel experiment (I am referring to the difference between Figs. S2a and S2b)? Surely the input polarization matters if one wants to address the correct optical mode in the OMC, and I can even imagine that there is some kind of polarization element before the filter cavities, such that once again a polarization controller would be needed.

-- The authors mention that an evanescent field coupling would be beneficial for future improvements on the front of single photon yield. Are the authors considering something along the line of their reference 35, or are they maybe referring to some three dimensional design where the evanescent coupling comes from the top? I am trying to understand how feasible this evanescent coupling is if one wants to ensure that the thermalization maintains the same quality, and that the fabrication does not become considerably more complex.

Version 1:

Reviewer comments:

Reviewer #2

(Remarks to the Author)

I have read carefully the replies of the authors, and I am thoroughly satisfied with their work. I do not have any further comments/concerns, and I recommend publication in Nature Communications.

We thank both reviewers for the thorough review of our work and their useful suggestions for improving our manuscript. We believe that through their comments the revised manuscript has improved significantly in quality. Below we address the referee comments point-by-point and explain changes made to the manuscript.

In addition to the changes made in response to referee comments, we implemented the following minor changes to the manuscript:

- Added a citation to a work that was recently published on the arXiv showing improved thermal performance of clamped optomechanical crystal structures (Kolvik, J., et al. "Optomechanical crystal in light-resilient quantum ground-state." arXiv:2510.15724 (2025).). The results of this work are highly relevant to the discussion of the results in our manuscript as they provide another path towards improved thermal noise performance and therefore improved single-photon purity.

Reviewer #1 (Remarks to the Author):

The paper demonstrates the performance of an efficient photon-phonon converter based on a 2D optomechanical crystal device already demonstrated recently in Ref. 34 (see also Ref 35). Similar 1D devices of this kind have been already employed for early demonstration of single photon manipulation, quantum teleportation and entanglement, but the purity of the generated single photon pulses was not enough for an efficient use in a quantum network. In this paper, the significantly improved management of the heating caused by optical absorption enables a much more pure single photon and single phonon states. This is very clearly demonstrated through a Hanbury Brown-Twiss experiment and then also a Hong-Ou-Mandel interference experiment, achieving a performance which is the best one achieved with optomechanical devices, and approaching the one achievable with atomic ensembles. I agree with the authors that such on-chip devices could be easily integrated within hybrid platforms with quantum dots and superconducting qubits, and therefore are extremely promising. The data are clearly presented and discussed, and the supplementary materials provide a quite complete description of the techniques and methodology.

For the above reasons I recommend publication in Nature Communication. A suggestion for a possible improvement is related to the physics behind the thermal management, which has been partially discussed in the previous works (Refs 34 and 35). Apart from the intuitive argument based on the more efficient 2D versus 1D geometry, could the author provide some more detail or a model of why now optical absorption is less effective? It is only a matter of better thermal conductance and dissipation? Which is the effect of optical power on the mechanical and optical quality factors? One could compare for example with the recent study of the mechanical quality factor versus T and optical power by F. Marzioni et al., Appl. Phys. Lett. 126, 174002 (2025), even though the latter refer to a 2D SiN membrane with a Si frame.

We would first like to thank the referee for their very positive assessment of our work and for supporting publication in Nature Communications. We also appreciate the suggestion for a discussion of the physics behind the optical absorption heating. A widely accepted phenomenological model has been established in earlier work on absorption heating in optomechanical crystal structures [5]: due to the existence of dangling bonds at the silicon/silicon dioxide interface, discrete interface defect states below and above the silicon electronic midgap can be formed due to unreacted Si dangling bonds or Si dangling bonds

interacting with oxygen atoms [1] [2] [3]. It is believed that upon absorption of the telecom photons through the interface states, a high-frequency (THz) thermal phonon bath is generated, which will go through three-phonon scattering processes involving the mechanical mode of interest and the local thermal bath, where the GHz phonons of frequency will be generated through the interaction between two high-frequency phonon modes [4] [5]. The general design strategy behind 2D OMC structure is to make use of the higher density phononic states resulted from a large contact area with the substrate, that can carry away heat before our mechanical mode of interest is populated [6]. In [6], the difference in thermal conductance between 1D nanobeam and 2D OMC structure has been simulated, and a factor of 42 enhancement in thermal conductance coefficient was expected for the quasi-2D OMC, which was believed to be the main reason for the improvement in thermal performance.

We have now added such a discussion to the Supplementary Information, section E, "Extended discussion of improved thermalization of 2D optomechanical crystal cavities".

Regarding the referee's question about the impact of changes in the optical quality factor at varying optical input power:

Aside from a shift of the cavity resonance, change in optical quality factor in response to different powers has been studied in [7], where different nonlinear effects (such as two-photon absorption and free carrier absorption) due to the high electromagnetic energy stored in OMC structure have been incorporated. When the power loaded to the cavity was increased to hundreds of μW , it will eventually lead to a broadening of the optical linewidth. However, even for the highest power loaded into the cavity used in our experiment ($\sim 1.6\text{fW}$), the change in optical quality factor is negligible according to the simulation in [7], which is consistent with our observations in the experiment. Moreover, considering the large difference in laser detuning ($\sim 10\text{GHz}$) and optical linewidth ($\sim 2.4\text{GHz}$), the resulted change from optical linewidth in scattering probabilities is even more negligible. Such a discussion has been added to the Supplementary Information, section D, "Measurement of thermal phonon occupancy from the readout pulse".

Regarding the referees question about the impact of changes of the mechanical quality factor at varying optical input power:

In previous work [6], there has been systematic investigations about how the mechanical linewidth changes with power inside the optical cavity for quasi-2D OMC structures. Similar to the 2D SiN membrane in [8] mentioned by the reviewer, higher power will decrease the mechanical quality factor also for quasi-2D OMC structure, e.g. mechanical decay rate increased from $\sim 2\pi \cdot 100\text{kHz}$ to $\sim 2\pi \cdot 160\text{kHz}$ when the intracavity photon number increases from 10^3 to 10^4 in [6] for quasi-2D OMC structure.

In our experiment, for each repetition cycle, $T_{\text{rep}} \gg$ mechanical life time, i.e. we wait long enough so the light-induced heat is completely dissipated and the device is fully thermalized to the cold environment ($\sim 20\text{mK}$) before the start of the next repetition cycle; therefore, the slightly deteriorated mechanical quality factor at higher optical power inside the cavity does not play a significant role in our experiment. Such a discussion has been added in Supplementary Information, section D, "Mechanical lifetime measurement".

Reviewer #2 (Remarks to the Author):

The authors present an experiment that involves a two-dimensional optomechanical crystal (OMC) operated in the pulsed regime of single phonons. They provide evidence for the single phonon nature of their mechanical state by performing a g^2 (Hanbury Brown-Twiss) experiment, and evidence for the indistinguishability of these single phonon from one run to another by performing a Hong-Ou-Mandel experiment. The work is excellently written and referenced, and the execution of the experiment is of the highest quality.

Two-dimensional adaptations of the original one-dimensional OMC is a crucial upgrade of this experimental platform, that may enable deeply quantum operations without the common obstacle of thermal heating originating from the control optics. The idea of extending the OMC to a two-dimensional configuration is not novel, and promising results have been already published recently by other groups. The focus of the authors on the single phonon nature and on the indistinguishability, however, elevates the novelty of the work and convinced me that it deserves publication in Nature Communication, which I recommend.

However, I would ask the authors to provide few clarifications of their work first.

-- For readers less familiar with fabrication intricacies, could the authors comment a bit more about the specific configuration of their device? Would it be possible to achieve the same effect with different unit cells instead of their stars and C shapes? Clearly yes, given the work in Nat. Comm. 16, 2576 (2025), so I wonder what are the advantages and disadvantages of different choices. Also, why is the waveguide that delivers light to the device curved? Is there something relevant on a physical level in such layout, or is it just for convenience?

We would like to thank the referee for their very positive assessment of our work and for recommending publication in Nature Communications. If we compare the snowflake + Cshape design with other 2D OMC designs such as the one in [9], the choice of boomerang unit cell in [9] was mainly motivated by bringing down the mechanical frequency (~ 7.4 GHz) to reduce the complexity of coupling to superconducting circuits. In addition, the boomerang unit cell has a larger filling factor (74.5%) than snowflake unit cells (64.1%), which might be beneficial for better thermalization. However, this is at the cost of smaller optical and mechanical bandgaps from the waveguide unit cell, which are ~ 27 THz wide for optical bandgap and ~ 0.15 GHz wide for mechanical bandgap, compared to ~ 32 THz wide optical and ~ 0.5 GHz wide mechanical bandgaps for the snowflake + Cshape unit cell. Moreover, the mechanical band of interest in [9] is very flat compared to snowflake + Cshape structure, which makes the former more susceptible to fabrication imperfections [6]. We have added a discussion of this design comparison to the Supplementary Information, section A, "Finite-element simulations".

The waveguide is curved by 45 degrees, because the dicing of the SOI chips is done for [110] crystal orientation, while the desired orientation of the device is along the [100] direction for a cleaner mechanical spectrum, as previously demonstrated in [11].

-- The thermal improvement reported by the authors is a bit lower than other two-dimensional OMC

experiments, could you estimate why is this the case? Was there a thermal analysis during the simulation stage?

We agree that the manuscript will offer more insight to the reader if we compare the thermal performance of different 2D OMC structures in more detail. For direct comparison, according to the plot in Fig.4 (c) in [9], with scattering probability of ~5%, and photon-phonon pair generation rate of 5 kHz, the estimated phonon occupation is between 0.02 to 0.03. For our Figure 2 (a), with scattering probability of ~5%, and the same photon-phonon pair generation rate of 5 kHz (since our repetition rate is 100kHz), the measured thermal phonon occupancy is 0.029, which is comparable to [9]. On the other hand, there can be some variation between devices for thermal performance due to fabrication imperfections.

For the side-coupled snowflake + Cshape structure investigated in [10], with a repetition rate of 250Hz, at 10% transduction efficiency, the phonon occupation is ~0.045; at 90% transduction efficiency, the phonon occupancy is ~0.25. For our end-coupled snowflake + Cshape structure, with a repetition rate of 100kHz, at 10% transduction efficiency, the phonon occupation is ~0.041; at 60% transduction efficiency, the phonon occupation is ~0.3. We can see that at high power and high transduction efficiency, the side-coupled structure in [10] does indeed perform better than our device used in the experiments (despite the different repetition rates used). This is mainly due to the side-coupled geometry that is used in [10], which mechanically detaches the optomechanical crystal cavity from the coupling waveguide which could act as a source of thermal noise [10]. The above comparisons have been added to the Supplementary Information, Section E, "Extended discussion of improved thermalization of 2D optomechanical crystal cavities".

The general design strategy behind 2D OMC structure is to make use of the higher density phononic states resulted from a large contact area with the substrate [6], that can carry away heat before the mechanical mode of interest is populated. In [6], the difference in thermal conductance between 1D nanobeam and 2D OMC structure has been simulated, and a factor of 42 enhancement in thermal conductance coefficient for the quasi-2D OMC was expected, which was believed to be the main reason for the improvement in thermal performance.

-- In the experimental setup, the authors seem to be using two different lasers for the red and for the blue sidebands. What are actually the powers involved in this experiment? Could they not have executed the same experiment with only one laser and some kind of single sideband modulator?

We agree that the manuscript would benefit from an explicit mention of optical powers the measurements are taken with. We added the pulse energy of the write and readout pulses for both the HBT and HOM experiments in the captions of Figs. 3 and 4 in the main text.

As correctly pointed out by the referee, we could have performed the same experiment by using only one laser and a single sideband modulator to generate a second, frequency-shifted optical beam. For the experiments presented in this work, this would work perfectly fine. The main reason for using two lasers instead of one + EOM is purely practical: the two lasers were already available in our lab at the time of the experiments and we did not have to make (the small) modifications of adding an additional EOM and filtering its sidebands.

-- Along the same line, I would have normally thought that the frequency of the lasers needs to be

stabilized by looking at their reflection from the optical resonator of the OMC. The authors instead seem to be using only a wavemeter, which makes their stabilization open-loop with respect to their device under test. Is this possible because of the relatively large linewidth of their resonator, or is the device really that stable?

As correctly pointed out by the referee, the laser stabilization is performed in an open-loop configuration with respect to the device resonance frequency. The reason why this is possible is that due to the relatively broad linewidth of the optical cavity resonance of our OMC ($\kappa/2\pi \sim 2.4$ GHz), drifts of the optical cavity resonance frequency is much smaller than the linewidth and will therefore not affect the induced optomechanical scattering probabilities. Compared to this linewidth, we do not observe any drifts of the optical resonance frequency over the whole measurement time of approximately three months. We agree that the reader would benefit from this additional information and we added a brief discussion to Section C of the Supplementary Information.

-- The authors report two devices with vastly different lifetimes, 1 μ s against 9ms. Was this result on purpose or accidental?

This result is on purpose by engineering the device geometry to have more or less phonon radiation loss. The device shown in Fig. S3(c) has an added air gap at the end of the device instead of a direct connection to the substrate for the device in in (b). This added air gap significantly reduces phonon radiation loss. We agree with the referee that this design choice can be explained in more detail in the manuscript. Therefore, we added a brief explanation in Section D of the Supplementary Information and added SEM images of the two devices presented in Fig. S3 to illustrate the differences in device geometry.

-- What role does the length and shape of your pulse play? Is there anything to gain from shaping it to a more tailored envelope, for instance?

The pulse shape of the optical pulses indeed plays an important role both in achieving optimal device performance and for future applications of our single photon generation scheme:

Firstly, to achieve minimal thermal noise we generally use the shortest possible optical pulses (~ 40 ns pulse duration as already stated in the manuscript) limited by the turn-on time of our acousto-optic modulator used for pulse generation (operation frequency 110 MHz). The optical absorption heating process has a characteristic time scale of few hundreds of nanoseconds as first identified in [12]. Therefore, longer optical pulses lead to an increased amount of added thermal noise. In our opinion, this aspect is already sufficiently explained in the second paragraph of the discussion section of the main text. We added the above mentioned reference both in the second paragraph of the discussion section as well as in the introduction of our manuscript when discussing the presence of thermal noise in optomechanical crystal structures.

Secondly, the pulse shape of the readout pulse determines the pulse shape of the emitted photons. This is important in future applications in which the bandwidth and more generally the pulse shape of the photons from our device needs to be matched to, for example, photons emitted from single quantum emitters such as rare-earth ions or quantum dots to be able to implement entanglement protocols based on single-photon interference with high fidelity. We added a sentence at the end of the second paragraph in the discussion section of our manuscript to discuss this.

-- In the pump and probe experiment of the Supplemental, the authors mention that their probe pulse is 5us long, and that it is shone on the SNSPDs. Why is the duration of this pulse so long if the temporal dynamics they try to resolve is on the order of few hundreds of nanoseconds (Fig. S4)?

We used a long pulse in order to capture the dynamics of the whole time range before knowing that the dynamics of interest appeared at a few hundreds of ns. We note that in this experiment the click probability of the SNSPD per trial is kept low ($p_{\text{click}} \ll 1$) by using a low probe photon number so that no distortion of the measurement data due to saturation effects of the SNSPD occurs despite the long pulse length.

-- Why did the authors remove the polarization controllers before and after the device for the Hong-Ou-Mandel experiment (I am referring to the difference between Figs. S2a and S2b)? Surely the input polarization matters if one wants to address the correct optical mode in the OMC, and I can even imagine that there is some kind of polarization element before the filter cavities, such that once again a polarization controller would be needed.

We thank the referee for finding this error in our illustration. The polarization controllers were indeed also present in the optical setup for the HOM experiment. We corrected this error in the updated version of Fig. S2.

-- The authors mention that an evanescent field coupling would be beneficial for future improvements on the front of single photon yield. Are the authors considering something along the line of their reference 35, or are they maybe referring to some three dimensional design where the evanescent coupling comes from the top? I am trying to understand how feasible this evanescent coupling is if one wants to ensure that the thermalization maintains the same quality, and that the fabrication does not become considerably more complex.

Indeed we are referring to the evanescent coupling geometry where the coupling is done from a coupling waveguide at the side of the OMC cavity as already successfully demonstrated in [10]. We believe that this is already sufficiently clarified in the manuscript by the reference to [10] in the text when we discuss the evanescent coupling in the main text. We appreciate the suggestion of the referee to explore alternative coupling geometries, such as evanescent coupling from the top. In fact, some of the authors of this paper have previously demonstrated fabrication techniques for such evanescent coupling from the top by using pick-and-place techniques of photonic structures [13]. These techniques could be implemented to achieve evanescent light coupling to 2D OMC from the top. We included a brief discussion of this approach including the above mentioned reference [13] in Section E of the Supplementary Information.

References:

[1] H. Kobayashi, Y. Yamashita, T.Mori, Y.Nakato, T. Komeda, and Y.Nishioka, Interface states for si-based mos devices with an ultrathin oxide layer: X-ray photoelectron spectroscopic measurements under biases, Jpn. J. Appl. Phys. 34, 959 (1995).

- [2] Y. Yamashita, K.Namba, Y.Nakato, Y.Nishioka, and H. Kobayashi, Spectroscopic observation of interface states of ultrathin silicon oxide, *J. Appl. Phys.* 79, 7051 (1996).
- [3] T. Sakurai and T. Sugano, Theory of continuously distributed trap states at si-sio₂ interfaces, *J. Appl. Phys.* 52, 2889 (1981).
- [4] G. P. Srivastava, *The physics of phonons* (CRC press, 2022).
- [5] S. M. Meenehan, J. D. Cohen, S. Groblacher, J. T. Hill, A. H. Safavi-Naeini, M. Aspelmeyer, and O. Painter, Silicon optomechanical crystal resonator at millikelvin temperatures, *Phys. Rev. A* 90, 011803 (2014).
- Change in mechanical Q-factor with power: mechanical Q is expected to deteriorate, but we use long enough repetition period anyway.
- [6] H. Ren, M. H. Matheny, G. S. MacCabe, J. Luo, H. Pfeifer, M. Mirhosseini, and O. Painter, Two-dimensional optomechanical crystal cavity with high quantum cooperativity, *Nat Commun* 11, 3373 (2020).
- [7] P. E. Barclay, K. Srinivasan, and O. Painter, Nonlinear response of silicon photonic crystal microresonators excited via an integrated waveguide and fiber taper, *Optics Express* 13, 801 (2005).
- [8] Marzioni, Francesco, et al. "Mechanical characterization of a membrane with an on-chip loss shield in a cryogenic environment." *Applied Physics Letters* 126.17 (2025).
- [9] Mayor, F. M., Malik, S., Primo, A. G., Gyger, S., Jiang, W., Alegre, T. P., & Safavi-Naeini, A. H. (2025). High photon-phonon pair generation rate in a two-dimensional optomechanical crystal. *Nature communications*, 16(1), 2576.
- [10] Sameer Sonar, Utku Hatipoglu, Srujan Meesala, David P. Lake, Hengjiang Ren, and Oskar Painter, "High-efficiency low-noise optomechanical crystal photon-phonon transducers," *Optica* 12, 99-104 (2025)
- [11] C. M. Kersul, R. Benevides, F. Moraes, G. H. M. De Aguiar, A. Wallucks, S. Gröblacher, G. S. Wiederhecker, and T. P. Mayer Alegre, Silicon anisotropy in a bi-dimensional optomechanical cavity, *APL Photonics* 8, 056112 (2023).
- [12] S. M. Meenehan, J. D. Cohen, G. S. MacCabe, F. Marsili, M. D. Shaw, and O. Painter, Pulsed Excitation Dynamics of an Optomechanical Crystal Resonator near Its Quantum Ground State of Motion, *Phys. Rev. X* 5, 041002(2015).
- [13] Guo, J., Gröblacher, S. Integrated optical-readout of a high-Q mechanical out-of-plane mode. *Light Sci Appl* 11, 282 (2022).